# KGMP: Augmenting retrieval knowledge graph with multi-hop perceptron

**Zhijie Yang**[1,2], **Liyuan Weng**[3], **Liang Zhang**[1,2], **Ruojia Tong**[1,2], **Jianyu Xie**[1], **Zhuo Zeng**[1], **Duanbing Chen**[1,2,4]*

**1** Big Data Research Center, University of Electronic Science and Technology of China, Chengdu, China, **2** Chengdu Union Big Data Tech. Inc., Chengdu, China, **3** Cuiying Middle School, Shantou, China, **4** Suining Institute of Digital Economy, Suining, China

* dbchen@uestc.edu.cn

**Data availability statement:** All datasets used in this paper are open source and can be

## Abstract

The core challenge of Knowledge Base Question Answering (KBQA), as a bridge between natural language and structured knowledge, is to accurately map complex semantic queries into Graph Query Language (GQL). Compared with the traditional Text-to-SQL task, KBQA faces a dual challenge: the structural differences between GQL and SQL and the lack of high-order subgraph information in multi-hop inference of knowledge graphs. While existing approaches such as ChatKBQA have made progress, the limitation of subgraph scalability severely constrains multi-hop query performance. To this end, this study proposes Knowledge Graph Multi-hop Perceptron (KGMP) - a retrieval-generation framework fine-tuned based on open-source large language models, whose innovativeness is reflected in three aspects: 1. **Dynamic Graph Traversal Mechanism:** Through an iterative subgraph expansion strategy, KGMP effectively achieves dynamic traversal of problem oriented graphs with progressive reasoning. 2. **Structured Interaction Protocol:** Based on SparQL syntax, KGMP designs a lightweight interaction instruction set to build an efficient communication interface between LLM and knowledge graph. 3. **Graph Structure Optimization Technique:** Develop subgraph reordering algorithms and pruning strategies based on the reranker model to ensure that the subgraphs input to the LLM are both compact and semantically complete. By integrating KGMP as a retrieval module into the ChatKBQA framework and providing it with optimised multi-hop subgraph input, the experimental results show a performance improvement of 6.2% and 5.3% on the WebQSP and CWQ datasets, respectively. This study provides a new technical paradigm for deep collaboration between LLM and knowledge graph.

## Introduction

The accelerated developments in the fields of finance, social networking and e-commerce have resulted in the formation of comprehensive and complex real-world information network. Nevertheless, conventional relational database management systems (DBMS) encounter considerable difficulties in processing and analysing graph-structured or network data. Consequently, graph-based databases are becoming increasingly prevalent as a means of efficiently

accessed via the following links:
WebQuestionSP: https://www.microsoft.com/
en-us/download/details.aspx?id=52763.
ComplexWebQuestions:
https://www.tau-nlp.sites.tau.ac.il/compwebq.

**Funding:** This work was partially supported by
the Key Research and Development Program of
Sichuan Province under Grant Number
2024YFG0008.

**Competing interests:** The authors have
declared that no competing interests exist.

storing and querying this essential information. Serving as a tool for the management of complex and diverse relationships, graph-based databases offer a practical and effective means of representing and storing data in a graphical format. However, due to the flexible and variable syntax of GQL, the learning curve of GQL is significantly steeper compared with SQL, underscoring the importance of KBQA.

The field of KBQA has traditionally addressed two fundamental challenges: knowledge retrieval [1] and semantic parsing [2]. The objective of knowledge retrieval is to identify relevant entities, relationships, or triples within the knowledge base. Semantic parsing, which may be conceived as a translation process, converts unstructured natural language questions into structured logical representations. These may be expressed in a variety of forms, including S-expressions [3] or executable GQLs such as Cypher Query Language (CQL). This enables the execution of queries on graph-based databases, thereby facilitating the retrieval of interpretable execution paths and precise answers.

Previous research on KBQA [4–6] have introduced a number of knowledge retrieval methods, including Named Entity Recognition (NER) [7], entity linking [8], and subgraph retrieval [4]. It is of great consequence that this retrieval step is conducted with the utmost precision and brevity, as the accuracy and succinctness of the retrieved results have a significant impact on the correctness of the semantic parsing outcomes. Once the relevant triples of facts have been retrieved, the researchers employ seq2seq models to directly derive the answers to the questions, as outlined in reference [9]. Furthermore, other scholars [10–12] have developed intermediate logical forms between the representation of natural language questions and the final answer. These forms convert and execute transformed graph-based database query languages to obtain answers. Additionally, some researchers [13] have combined the aforementioned methods to further enhance the accuracy of the KBQA.

Nevertheless, the existing KBQA methodology still faces some significant challenges, as shown in the upper part of Fig 1. The lack of high-order subgraph data makes pattern queries more difficult. The generation of structured query statements from natural language queries alone results in a large number of false candidates, which significantly increases the time complexity of retrieval. Furthermore, relying solely on natural language queries to construct structured queries is insufficient to ensure their accuracy. In multi-hop question answering, high-order subgraph information is essential for the accuracy of KBQA. In the absence of this information, semantic parsing may lack adequate pattern references.

In order to address these challenges, we introduce KGMP, a novel retrieval-generation KBQA framework that makes use of fine-tuned open-source LLM libraries, including LLaMA-2-7B [14] and Qwen-7B [15]. As shown in the lower part of Fig 1, KGMP employs an intuitive approach. Initially, it extracts entities and then queries the graph-based database for triples associated with the current entity. In the event that CQL cannot be generated based on the available information, higher-order subgraphs will be queried. During the generation phase, open-source LLMs are fine-tuned with instruction-tuning techniques [16] to equip them with multi-hop reasoning and CQL format generation capabilities.

The main contributions of the paper are summarized as:

- An information retrieval method based on LLMs named KGMP is proposed for KBQA tasks.
- KGMP can autonomously traverse KG, filling the gap of the current KBQA method lacking in high-order subgraphs.
- The performance of KGMP was evaluated on CWQ and WebQSP. The accuracy was improved significantly by integrating KGMP.

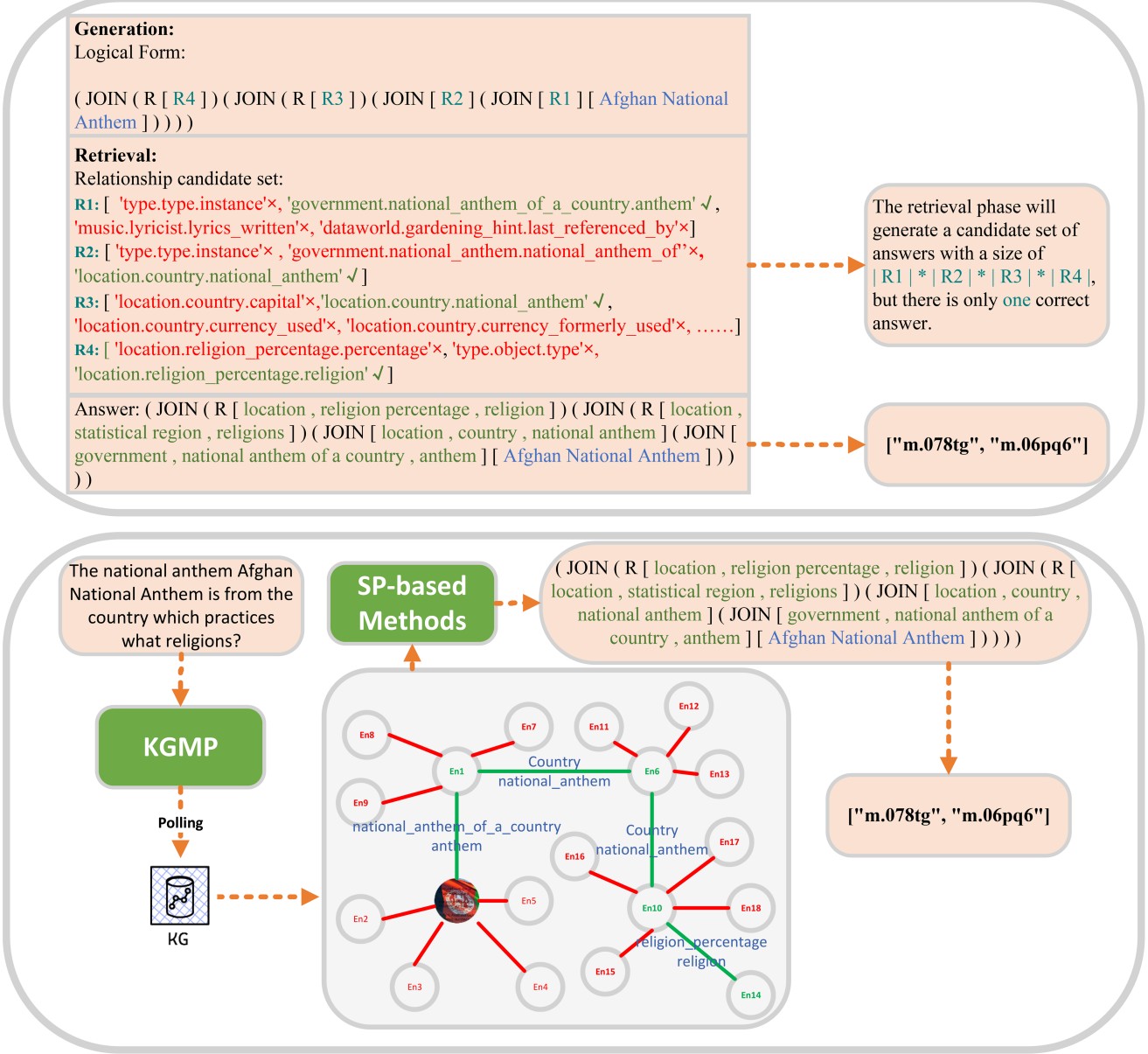

**Fig 1. Example of comparison between previous retrieval methods (top) and our proposed KGMP (bottom).**

## Related works

### KBQA methods

The existing KBQA methods can be broadly classified into two categories: information retrieval (IR) and semantic parsing (SP). IR-based methods aim to retrieve relevant triples

or subgraphs from the knowledge graph to infer answers for a given query. For example, GRAFT-NET [17] extracts answers from question-related subgraphs and linked text; Sun [18] constructs memory through a key-value memory network to address issues such as the limitations of knowledge bases in KBQA, patterns of storing knowledge that may not support certain types of answers, and the sparsity of knowledge bases. The advent of novel mechanisms to enhance subgraph-based KBQA has been marked by frameworks such as TransferNet [19], Relation Learning [20], Subgraph Retrieval [4], Unik-QA [5], and SKP [21] have introduced novel mechanisms to enhance subgraph-based KBQA.

SP-based methods tackle KBQA by converting natural language questions into GQL or logical forms, typically involving two steps: (1) transforming the question into an executable semantic representation using a semantic analyzer, and (2) using this representation to query the knowledge graph and retrieve the answers. Since the correctness of the semantic representation directly affects the correctness of the answer, the core task of this type of question-answering method lies in the semantic analysis task, i.e. how to generate the semantic representation corresponding to natural language questions. Some SP-based methods, such as Rigel [22], UniKGQA [23], BeamQA [24], HGNet [25], StructGPT [6] and PanGu [26] use multi-step query graphs and search strategies for semantic parsing. Additionally, some methods such as ReTraCk [27], CBR-KBQA [28], RnG-kbqa [10], Program Transfer [29], TIARA [12], ArcaneQA [30], GMTKBQA [11], Uni-Parser [31], Unified-SKG [32], DECAF [13] and FC-KBQA [33] use sequence-to-sequence models to generate s-expressions completely and provide various enhancements to the semantic parsing process. ToG-2 [34] aims to achieve a deep and comprehensive retrieval process by iteratively extracting information from both unstructured text and structured knowledge graphs. In contrast, Xu et al. [35] model the knowledge graph as a tree for question answering. Meanwhile, OneGen [36] emphasizes the seamless integration of generation and retrieval.

### LLM

Large language models (LLMs), characterized by extensive parameter sizes and abundant training data, have become powerful tools in natural language processing. Notable examples of LLMs include transformer-based models such as OpenAI's GPT series [37] and Google's BERT [7]. These models are capable of acquiring profound language knowledge from text corpora through large-scale unsupervised learning, equipping them with robust language understanding and generation capabilities. Through the processes of encoding, generating, and decoding input text, LLMs excel in tasks such as semantic understanding, automatic summarization, and question answering, thereby driving significant advancements in natural language processing. The integration of LLMs with KBQA can harness the respective strengths of both approaches, leading to enhanced language comprehension and reasoning.

**Knowledge enhancement.** LLMs can utilize knowledge graphs to enhance their comprehension of semantic relationships and contextual information. Techniques such as instruction tuning [38], LoRA [39], and QLoRA [40] have significantly augmented LLMs' domain-specific capabilities. Fine-tuning on domain-specific knowledge graphs enables LLMs to capture more accurate semantic relationships between entities and contextual information, thereby enhancing their performance in tasks such as question answering and summarization.

**Knowledge update.** It is often necessary for LLMs to update their knowledge on an ongoing basis in order to adapt to new scenarios and requirements. The updating of traditional knowledge often necessitates considerable manual input and expertise, resulting in prolonged cycles. In contrast, the utilisation of knowledge graphs for knowledge updating can be more efficient and automated. When updating knowledge within a knowledge graph, only the data

layer requires modification, and the LLMs do not require re-fine-tuning. The LLMs can continue to employ the original IR and SP methods to interact with the updated database and complete the question-answering task.

**Hallucinations.** In the natural language understanding and reasoning tasks, LLMs are prone to encountering hallucinations [41], which can give rise to misjudgements on the part of the model or erroneous outcomes in reasoning. The integration of knowledge graph representations with LLMs has been demonstrated to be an effective means of reducing hallucinations, thereby enhancing model accuracy in complex scenarios, thereby improving the model's efficacy in intricate settings [42].

## Method

### Definition

The objective of this study is to develop a model that translates natural language queries into executable graph queries for a graph-based database, as depicted in Fig 2. The transformation process can be formalised as follows:

$$c = f(q, \mathcal{G}_{en}^n), \tag{1}$$

where the variable *en* represents entities within the query, $\mathcal{G}$ denotes the graph based database, and $\mathcal{G}_{en}^n$ refers to an *n*-hop subgraph centered around *en*. The user input *q* represents a natural language query, while *c* denotes the generated executable GQL query. The database structure is defined as a set of triples $\mathcal{G} = \{(h, r, t) | h, t \in \mathcal{V}, r \in \mathcal{E}\}$, where $\mathcal{V}$ is the node set and $\mathcal{E}$ represents the set of relationships (edges).

### Overview of KGMP

KGMP is a specialized retrieval-generation KBQA framework tailored for KBQA, utilizing fine-tuning open-source LLMs. Firstly, The framework constructs a new dataset derived from the KBQA dataset, containing natural language queries, GQL pairs, and corresponding historical query dialogues, to facilitate efficient fine-tuning of LLMs. Upon encountering an entity *en* in a query, the system fetches the first-order subgraph $\mathcal{G}_{en}^1$ from the knowledge graph $\mathcal{G}$, relevant to the entity. Then, this subgraph is reranked and pruned to obtain $\mathcal{G}_{en}^{1'}$. $\mathcal{G}_{en}^{1'}$ is subsequently fed into the fine-tuned LLMs for analysis, where they determine if the current subgraph information is sufficient to generate an executable GQL. If the model deems the GQL unattainable from the current subgraph, it will either directly expand to higher-order subgraphs or select an edge *r* ($r \in \mathcal{G}$) to further expand to higher-order subgraphs, until a response is derived based on the available information. The basic structure is shown in Fig 2. After the traversal concludes, KGMP outputs its traversal path and the pruned results, both of which serve as inputs to the SP-based model for predicting the final graph query language.

To perform the Q&A task, ChatKBQA first converts the input natural language query into a logical expression skeleton. It then fills this skeleton with specific content based on similarity metrics. However, when creating the logical skeleton, ChatKBQA only uses the query itself to estimate the hop count of relevant knowledge graph subgraphs, lacking detailed subgraph information. To overcome this limitation, KGMP provides the retrieved knowledge graph subgraphs and their retrieval processes to ChatKBQA as historical dialogue context. This enhances ChatKBQA's understanding of subgraph structures, thereby improving the accuracy and completeness of answers.

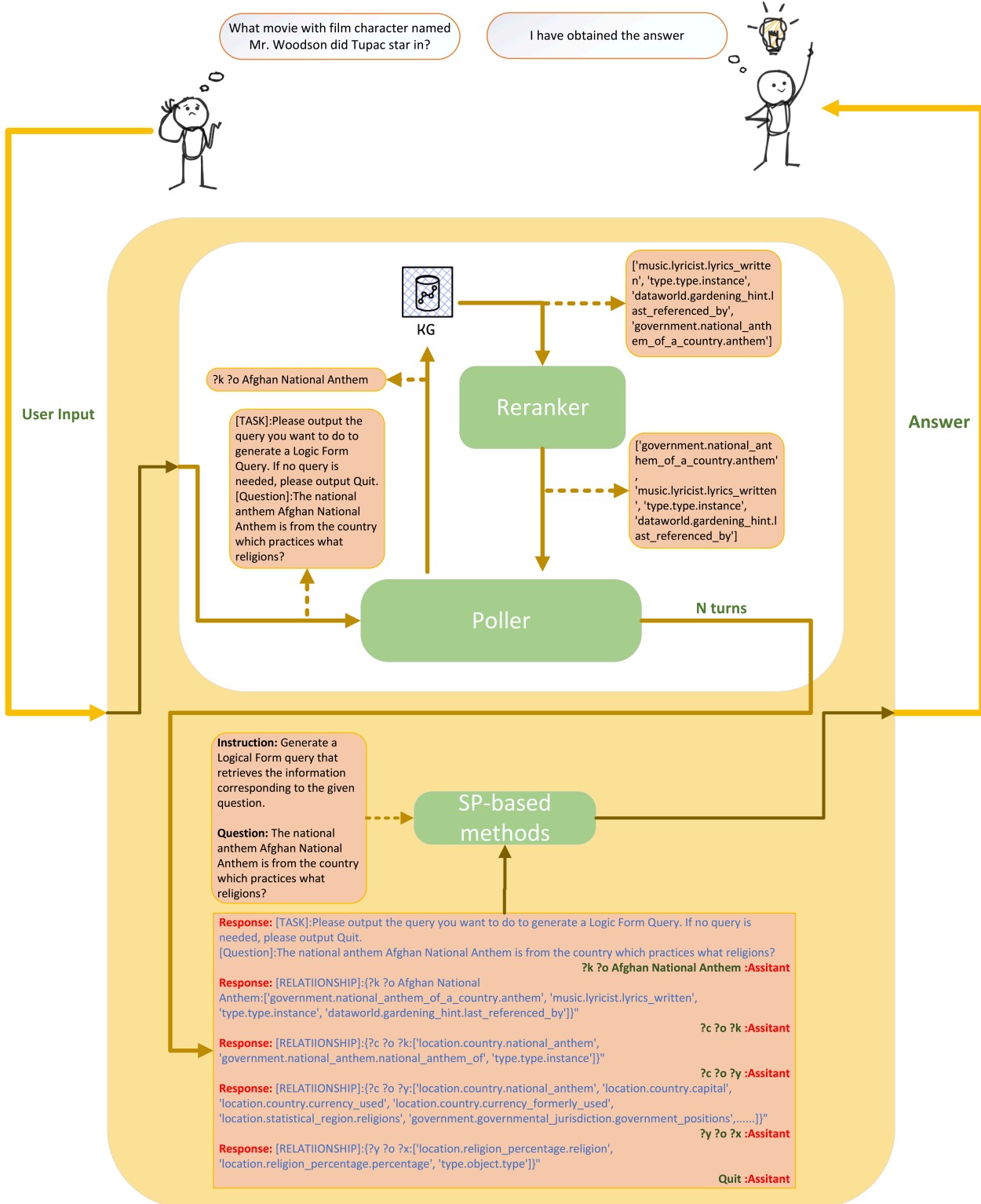

**Fig 2. An overview of KGMP: a KBQA framework leveraging fine-tuned LLMs with a retrieval-first approach before query generation.**

## Reranker

Under the KGMP framework, the process of converting natural language queries into structured logical representations or GQL can be divided into three stages: (1) **Selecting an appropriately sized subgraph:** This step is of great consequence, as it provides indispensable assistance for the subsequent semantic parsing stage. (2) **Ranking and pruning the subgraph:** Liu et al. [43] pointed out that LLM exhibits superior performance when the context is shorter and the answer is presented earlier in the context. In order to enable the model to obtain answers more effectively, the KGMP framework employs a ranking and pruning process for the subgraph. This step is of pivotal importance in ensuring the quality and relevance of the answers produced during the semantic parsing stage. (3) **Inputting the obtained subgraph and question into the model** to generate a structured logical representation.

To address the subgraph pruning problem in stage (2), we propose a preliminary model to rerank and prune the subgraph. The task is defined as:

$$\mathcal{G}_{en}^{n'} = \text{reranker}(\mathcal{G}_{en}^{n}, q, m), \tag{2}$$

where $\mathcal{G}_{en}^{n}$ represents the $n$-th order subgraph of entity $en$, $\mathcal{G}_{en}^{n'}$ represents the subgraph reranked and pruned according to query $q$. The reranking method may employ a variety of models, including pretrained models such as SimCSE [44] and LLMs or traditional algorithms such as Levenshtein distance. $m$ represents the maximum size of the subgraph. The reranker ensures that during the polling process between LLMs and the knowledge graph, the subgraphs obtained by the LLMs are closely related to the query.

In particular, the KGMP algorithm initiates the process by serializing the subgraphs. This involves encoding the entity "en" and its first-order subgraph as multiple one-to-many triplet representations, such as: $\langle en, r1, [t_{r1}^1, t_{r2}^2, ...] \rangle$, $\langle en, r2, [t_{r2}^1, t_{r2}^2, ...] \rangle$, etc. For preliminary pruning, reranker calculate the similarity between the natural language query and the relations $r$, sort the similarities in descending order, and retain the top $k$ relationships. To refine the triplets further, KGMP decomposes the preliminarily pruned one-to-many triplets into one-to-one triplet representations, such as: $\langle en, r1, t_{r1}^1 \rangle$, $\langle en, r1, t_{r1}^2 \rangle$, ..., $\langle en, r2, t_{r2}^1 \rangle$, $\langle en, r2, t_{r2}^2 \rangle$, .... KGMP further use the reranker to calculate the similarity between the natural language query and the complete triplets (including the tail entities) to evaluate the relevance of the tail entities with the natural language query. The similarities are sorted in descending order, and for each relation $r$, KGMP retain the top $p$ tail entities. Finally, the fully pruned triplets are re-represented as $k$ one-to-many triplet texts.

## Poller

The poller assesses the sufficiency of the current subgraph for the LLM to complete the KBQA. At the outset, a protocol is established whereby the LLM exits the polling process if the subgraph contains sufficient information. Otherwise, the LLM will select an entity and the direction of its connected edge to query higher-order subgraphs. The KG then executes the query and passes the result to the ranker for reranking and pruning. To minimize errors during polling, the use of simplified queries is strongly encouraged.

In order to reduce the likelihood of errors, the KGMP only requires the LLM to issue concise commands during the polling phase to interact with the knowledge graph. KGMP defines three types of interaction commands: (1) "en ?o ?y" is used to query all triplets where $en$ is the head entity. (2) "?y ?o en" indicates querying all triplets where $en$ is the tail entity. To enable the LLM to further query higher-order subgraphs, KGMP uses a symbol mapping to store the results of the queried triplets. For example, the query result for "Emmitt Smith ?o ?y" would

be stored as "Emmitt Smith ?o ?y": result. (3) The last type of interaction command is for querying previously appeared proxy entities. For instance, if a proxy entity "?y" appears in a previous query, KGMP can then perform further queries on that entity. An example would be "?y ?o ?x", which means using the result of the "Emmitt Smith ?o ?y" query as the head entity to query all relevant triplets. KGMP saves all polling conversation information and evaluates whether it can answer the natural language query based on the historical information.

### Efficient fine-tuning on LLM

In order to construct training data for fine-tuning instruction, it is first necessary to extract entities from the training set of the KBQA dataset and to determine the number of hops for the GQL that is related to the current query. The next step involves simulating LLM interactions with the database in accordance with the aforementioned calculations. Take the question "what electorate does Anna Bligh represent?" as an example, the golden entity in the question is "Anna Bligh", and the corresponding ID of this entity in the database is "m.072_m3", so the corresponding SPARQL query for this question is:

```
SELECT DISTINCT ?x WHERE {
m.072_m3 government.politician.government_positions_held ?y .
?y government.government_position_held.district_represented ?x .
}
```

Further extracting the critical path triples in SPARQL:

$$\langle m.072\_m3, government.politician.government\_positions\_held, ?y \rangle$$
$$\langle ?y, government.government\_position\_held.district\_represented, ?x \rangle$$

At this stage, the training data for simulating human experts can be constructed based on the two existing triples. First, a subgraph traversal is performed for the entity ID "m.072_m3", which acts as the head entity in the triples. KGMP queries the candidate relationships $r$ associated with this entity in the knowledge graph with this entity as the head entity, and sort and prune $r$. After that, KGMP uses the relationship of the first ternary as the golden relationship to select the direction in the subgraph of the entity "m.072_m3", and traverses the higher-order subgraphs in accordance with this relationship. In this process, "?y" represents intermediate entities, and KGMP does not concern itself with the specific nature of "?y", e.g.,

$$\langle m.072\_m3, government.politician.government\_positions\_held, ?y \rangle$$

Here, "?y" represents a set of entities related to the head entity "m.072_m3" via the relationship "government.politician.government_positions_held". Observing that in the second triple, "?y" occupies the head entity position, KGMP continues the traversal with "?y" as the new head entity, proceeding to explore higher-order subgraphs. This process repeats until the critical path triples in SPARQL are fully traversed. Once the traversal is complete, a command message is generated to instruct the LLM to produce the final GQL based on the problem and subgraph state. An example is shown in Fig 3.

Specifically, the inputs include the natural language query $q$, the set of entities *entities*, the mapping function $f$ for the fine-tuned LLM, the graph database $\mathcal{G}$, and $n$ is the maximum size of the subgraph. Initially, a *history* list is defined, and the current subgraph is recorded

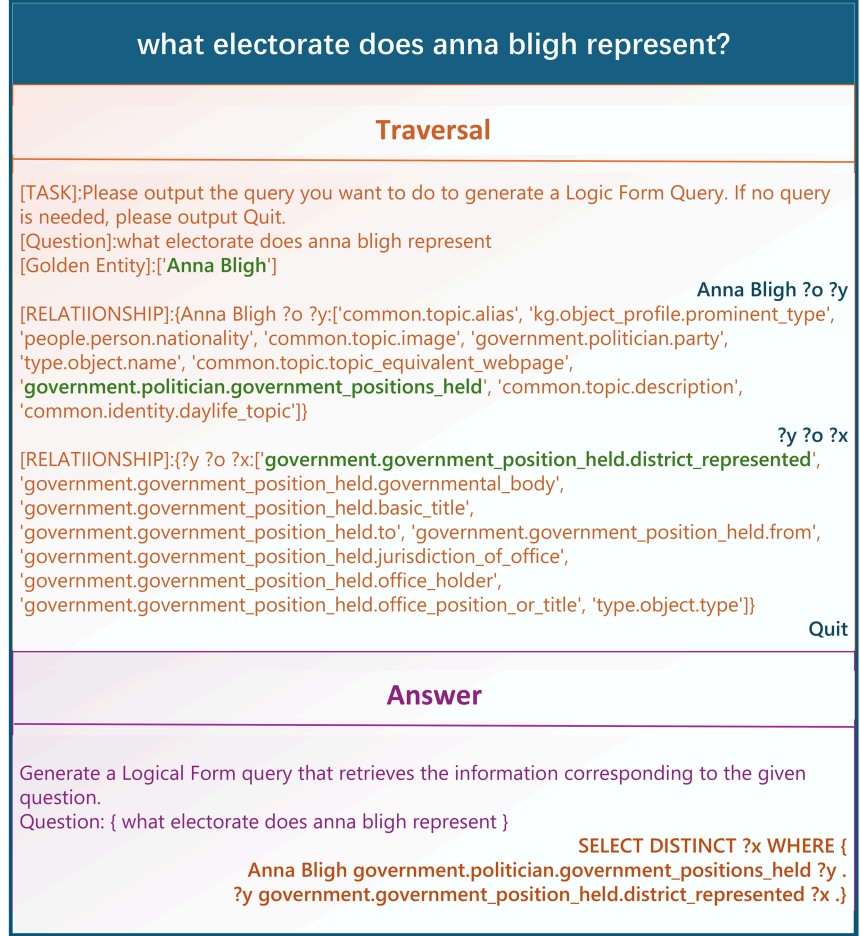

**Fig 3. Example of training data.** Each piece of training data is divided into two parts, traversal and answer, with the data on the left side representing instructions or database query results, and the data on the right side representing the KGMP outputs.

using *hop*. Then, querying the information of the *hop*-th order subgraph for entities in *entities* and saving it to *subgraph*. Subsequently, *subgraph* and *q* are input into the ranker for sorting and pruning, retaining up to *n* entries at most. Next, the sorted *subgraph* is input into the LLM, and *history* is updated. Finally, the response of the LLM is checked. If not, higher-order subgraphs are traversed based on the entities and directions in the *response*; otherwise, the obtained subgraph from polling and the question are input into the LLM to generate an *answer*.

By employing the retrieved data, LLM can augment its comprehension of the pivotal information essential for generating the present natural language query, obviating the necessity for trial and error. In this process LLM will learn two key competencies, (1) to determine whether the subgraph order is satisfied based on the problem; (2) to make the right direction for dynamic subgraph expansion based on the problem and historical information. The GQL process should be conducted within the dataset by replacing the obscure entity IDs with entity labels, thus facilitating a more comprehensive comprehension of the entity data by large models. Subsequently, the natural language questions and GQL or corresponding logical forms

should be treated as "inputs" and "outputs", respectively, with the queried data serving as "history". The "instruction" for this process is *Generate a Logical Form query that retrieves the information corresponding to the given question.* These elements constitute the instruction fine-tuning training data for open-source LLM.

Consequently, the integration of LLM with KG provides a robust and interpretable methodology for the generation of graph queries for KBQA tasks.

## Results and discussion

### Settings

**Datasets.** The experiments were conducted using two standard KBQA datasets. the WebQuestionSP dataset (WebQSP [45]) comprises 4,737 natural language questions with corresponding SPARQL queries, while the ComplexWebQuestions (CWQ [46]) includes 34,689 natural language questions with corresponding SPARQL queries. Both datasets were constructed using Freebase KB by Zhang et al. [47].

**Baselines.** In this paper, we compared 25 baselines, including: Smaller Language Models Fine-Tuned [12,30], LLM ICL [6,48], and LLM Fine-Tuned [49]. All the compared baselines are listed in Table 1.

**Evaluation metrics.** Three evaluation metrics, F1 score, Hits@1, and Accuracy (Acc) are employed to assess the comprehensiveness of all answers, the top-ranked single answer, and the exact match accuracy on WebQSP and CWQ, respectively.

**Hyperparameters and Environment.** The hyperparameters and experimental setup are as follows: the base model used by the reranker is bge-reranker [50]. the LLM is fine-tuned 100 times on the WebQSP dataset and 10 times on the CWQ dataset. The batch size is set to 32, with a learning rate of 5e-5, utilizing the LoRA fine-tuning method. All experiments are performed on a single NVIDIA H800 GPU with 80GB memory.

### Main results

In the KBQA, we employ the KGMP as a plug-and-play module, with the results integrated into ChatKBQA, as illustrated in Table 1. The configuration of ChatKBQA is as follows: for WebQSP, LLaMA-2-7B is fine-tuned using the LoRA method without beam search; for CWQ, LLaMA-2-13B is similarly fine-tuned. During the unsupervised phase, SimCSE [44] is employed for similarity calculation. The results demonstrate that incorporating KGMP leads to a notable enhancement in comparison to existing KBQA methods. The use of golden entities, the F1 score, Hits@1 and Accuracy on WebQSP increased by 2.8%, 0.7%, and 6.2%, respectively, compared to the previous best result, ChatKBQA; on CWQ, they increased by 4.8%, 1.9%, and 5.3%, respectively. This suggests that ChatKBQA has achieved state-of-the-art performance in KBQA after incorporating KGMP.

Although ChatKBQA is capable of capturing the structural features of a specific knowledge graph (KG) through fine-tuning, it is unable to accurately identify and locate target triples, particularly in complex, multi-hop reasoning scenarios. In contrast, the integration of KGMP establishes a retrieval-generation question-answering approach. The integration of KGMP provides precise and concise answer support for ChatKBQA, enabling the LLM to progress methodically towards the final answer, with each step closely linked. This integration offers two notable advantages. Firstly, the plug-and-play feature of KGMP allows seamless integration with other SP methods, thus enhancing the IR stage. Secondly, KGMP provides accurate factual triples for SP methods, empowering LLM to not only generate KBQA answers based

**Table 1. Comparison of StructGPT+KGMP and ChatKBQA+KGMP with other benchmarks on the WebQSP and CWQ datasets.**

| Model | WebQSP | | | CWQ | | |
|---|---|---|---|---|---|---|
| | F1 | Hits@1 | Acc | F1 | Hits@1 | Acc |
| **Smaller Language Models Fine-Tuned** | | | | | | |
| Subgraph Retrieval* [4] | 64.1 | 69.5 | - | 47.1 | 50.2 | - |
| TransferNet [19] | - | 71.4 | - | - | 48.6 | - |
| Relation Learning [20] | 64.5 | 72.9 | - | - | - | - |
| CBR-KBQA [28] | 72.8 | - | 69.9 | 70.0 | 70.4 | 67.1 |
| Rigel* [22] | - | 73.3 | - | - | 48.7 | - |
| BeamQA* [24] | - | 73.4 | - | - | - | - |
| UniKGQA* [23] | 72.2 | 77.2 | - | 49.4 | 51.2 | - |
| UnifiedSKG [32] | 73.9 | - | - | 68.8 | - | - |
| ReTraCk* [27] | 74.7 | 74.6 | - | - | - | - |
| RnG-KBQA [10] | 75.6 | - | 71.1 | - | - | - |
| ArcaneQA [30] | 75.6 | - | - | - | - | - |
| Uni-Parser* [31] | 75.8 | - | 71.4 | - | - | - |
| Program Transfer* [29] | 76.5 | 74.6 | - | 58.7 | 58.1 | - |
| HGNet* [25] | 76.6 | 76.9 | 70.7 | 68.5 | 68.9 | 57.8 |
| GMT-KBQA [11] | 76.6 | - | 73.1 | 77.0 | - | 72.2 |
| TIARA* [12] | 78.9 | 75.2 | - | - | - | - |
| SKP [21] | - | 79.6 | - | - | - | - |
| DECAF [13] | 78.8 | 82.1 | - | - | 70.4 | - |
| UniK-QA [5] | 79.1 | - | - | - | - | - |
| FC-KBQA [33] | 76.9 | - | - | 56.4 | - | - |
| **LLM ICL** | | | | | | |
| StructGPT* [6] | - | 72.6 | - | - | - | - |
| StructGPT+KGMP* **(ours)** | - | **83.6** | - | - | - | - |
| ChatGPT* [51] | 61.2 | - | - | 64.0 | - | - |
| FiDeLis [48] | 78.3 | 84.4 | - | 64.3 | 71.5 | - |
| PanGu [26] | 79.6 | - | - | - | - | - |
| **LLM Fine-Tuned** | | | | | | |
| ChatKBQA* [49] | 83.5 | 86.4 | 77.8 | 81.3 | 86.0 | 76.8 |
| ChatKBQA+KGMP* **(ours)** | **86.3** | **87.7** | **84.0** | **86.1** | **87.9** | **84.1** |

* Denotes using oracle entity linking annotations.

on subgraph information from KGMP but also to systematically approach the answer through the polling process, thereby completing the task.

Furthermore, it was observed that under in-context learning (ICL), LLM is capable of exhibiting performance that is comparable to that of fine-tuned smaller language models. This is because LLM has been extensively exposed to GQL during the pre-training phase and has acquired substantial related knowledge. This allows them to effectively recall and reuse this previously acquired knowledge in ICL. Nevertheless, the performance of ICL with LLM remains somewhat inferior to that of the fine-tuning strategy. The reason for this is that during the pre-training phase, LLM absorbs vast amounts of knowledge across multiple domains. This is done with the primary design goal of achieving cross-domain generalization. Therefore, when faced with specialized tasks in a specific domain, LLM requires additional learning in order to deepen their understanding and achieve optimal performance. This highlights that despite their strong foundational capabilities, LLMs require targeted fine-tuning in order to optimize performance in specialized domains.

## Validity analysis

In order to ascertain the efficacy of the KGMP framework, we proceeded to integrate KGMP with StructGPT and subsequently conducted tests on the WebQSP. The evaluation metric was Hits@1. The experimental results demonstrate that the incorporation of KGMP with Struct-GPT on WebQSP enhances Hits@1 from 72.6 to 83.6. The specific results are presented in Table 1. A comparison of the Hits@1 metrics reveals that the combination of StructGPT and KGMP resulted in an improvement of 11 points, while the combination of ChatKBQA and KGMP resulted in an improvement of 1.3 points. It is postulated that the marked enhancement in efficacy subsequent to the integration of KGMP in the former is attributable to the intrinsic capabilities of the agent model. The KGMP task is typically divided into two phases. The initial step is to issue a query pertaining to the subgraph in question. This process necessitates that the model possess two distinct capabilities. The first is the capacity to evaluate the sufficiency of the information provided by the current subgraph. The second is the ability to make decisions regarding the direction in which the higher-order subgraph should be expanded. These two capabilities align closely with the requirements for the LLM in the agent. The second step is to generate an executable GQL based on the existing subgraph information and the model's capabilities. This step necessitates a model with robust reasoning capabilities. The Open Compass leaderboard can be accessed via the following link Open Compass. The agent scores for GPT3.5-turbo and LLaMA-2-7B-Chat are 73.5 and 25.2, respectively, while the reasoning ability scores are 21.2 and 15, respectively. The data clearly demonstrate that GPT3.5-turbo exhibits superior performance in terms of judgment, decision-making, and reasoning compared to LLaMA-2-7B-Chat. Specifically within the KGMP framework, when confronted with an identical context, GPT3.5-turbo is more adept at discerning whether a more intricate subgraph is necessary for comprehension, and also displays enhanced accuracy in causal relationship reasoning.

## Error analysis

To systematically analyze the changes of StructGPT before and after adding KGMP, we randomly selected 100 error cases from the WebQSP. Subsequently, we manually inspected these errors and followed the classification method of Jiang et al. [6], dividing them into five categories.

- **Selection Error:** The LLM was unable to correctly identify the pertinent information.
- **Reasoning Error:** After presenting the extracted relevant information, the LLM was unable to generate a practical answer or GQL.
- **Generation Format Error:** The generated answer had an abnormal format and could not be recognized by our result parser.
- **Hallucination:** The generated result did not match the extracted information.
- **Other Errors:** Miscellaneous errors that do not fall into the above categories.

The relevant statistics are presented in Fig 4. It is evident that the incorporation of KGMP resulted in a notable reduction in the prevalence of selection errors, from the initial 74% to 44%. Liu et al. [43] pointed out that the ability of the LLM is related to the length of the input text. However, when KGMP is confronted with intricate queries, it frequently yields high-order subgraphs for the LLM, and the length of the input text is typically longer. Nevertheless, it attains superior outcomes, which is at odds with the assertion of Liu et al. [43]. We speculate that Rerank in KGMP plays a key role.

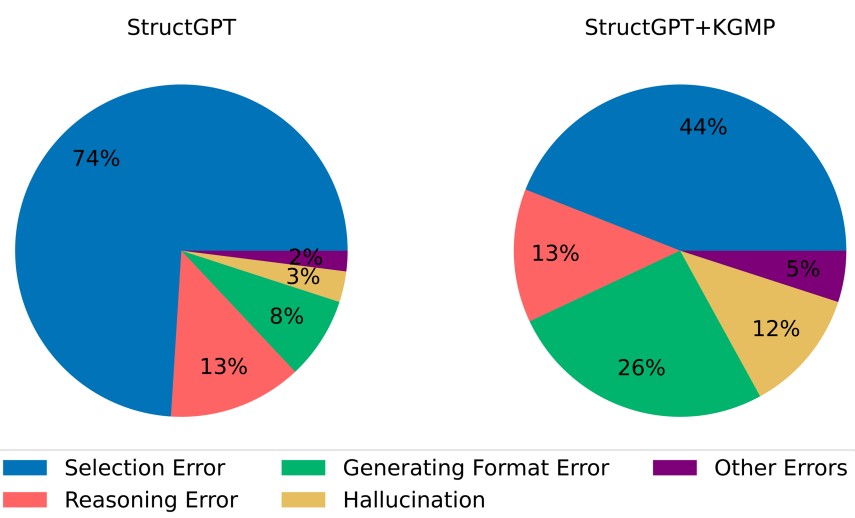

**Fig 4. Comparison of the proportion of different error types on WebQSP before and after joining KGMP.**

In order to ascertain the efficacy of the Reranker, an attempt was made to remove the Reranker from KGMP. The results are presented in Fig 5. The experimental results demonstrate that the reranker is a pivotal component of the KGMP. The removal of the Reranker resulted in a significant decline in the efficacy of ChatKBQA + KGMP. The context length of LLaMA is 4096. In WebQSP, 47.71% of the data subgraphs exceed this length, while in the CWQ dataset with more abundant multi-hop queries, this proportion reaches 73.49%. If only the multi-hop accuracy of predictions that do not exceed the context length is counted, the accuracy is also much lower than that of the complete ChatKBQA + KGMP. Combining the conclusion of Liu et al. [43], we speculate that providing the entire subgraph to the LLM will make it difficult to determine the answer among the huge candidate set. At the same time, KGMP introduces more noise rather than useful information.

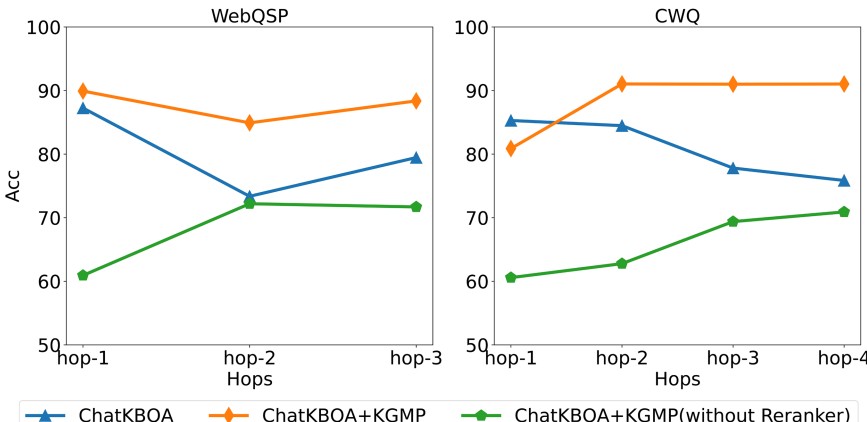

**Fig 5. Comparison of LLM accuracy in predicting multi-hop queries. The results of ChatKBQA+KGMP(without Reranker) only counted data with subgraph sizes not exceeding 4096 Tokens, which accounts for, only 53.29% of WebQSP data and 26.51% of CWQ data were counted.**

## Limitations analysis

Empirical evaluations have demonstrated that integrating the KGMP module into the ChatKBQA framework can substantially improve performance on standard KBQA datasets (WebQSP and CWQ), thereby enhancing the overall robustness and flexibility of the architecture. Nevertheless, despite these promising results, several limitations of KGMP persist. To examine these constraints, this section analyzes KGMP's computational overhead and accuracy variations with long-sequence inputs, its performance on ultra-large-scale knowledge graphs, and its reasoning capability in the absence of higher-order subgraphs.

**Model input length.** In the retrieval phase, KGMP necessitates the completion of several rounds of question-answering activities to construct the knowledge graph structure. These multiple interactions result in a notable increase in the input token length of the model. As illustrated in Fig 6, which contrasts the input token lengths for the ChatKBQA and ChatKBQA+KGMP frameworks, the incorporation of the KGMP module results in a notable expansion in the average token length. In particular, the average input token length for the CWQ dataset increased from 36.8 to 463.0, while for the WebQSP dataset it increased from 28.1 to 339.0. Furthermore, we examined the mean number of dialogue turns following the incorporation of the KGMP module. Our findings revealed that, in comparison to the ChatKBQA framework, the mean number of dialogue turns increased from one turn in the CWQ dataset to 4.8 turns, and from one turn in the WebQSP dataset to 3.6 turns, respectively. While augmenting the number of dialogue turns can enhance the model's comprehension of the graph, it also considerably extends the input length, resulting in a substantial escalation in computation time and storage necessities.

**Floating-point operations.** It is evident that an increase in the length of input tokens will inevitably lead to a proportional increase in the demand for computing power. Consequently, we undertake an analysis of the rise in FLOPs that can be attributed to the use of longer inputs. In order to calculate FLOPs, it is necessary to take into account both the self-attention mechanism and the feed-forward neural network (FFN) components of the base model, which in this case is LLaMA2-7B. The LLaMA2-7B model has 32 layers (L), with a hidden layer dimension $d_{model} = 4096$, a FFN dimension $d_{ff}$ four times that of the hidden layer (i.e. 16384), 32 attentional heads (H), and an individual head dimension $d_k = d_{model}/H = 128$.

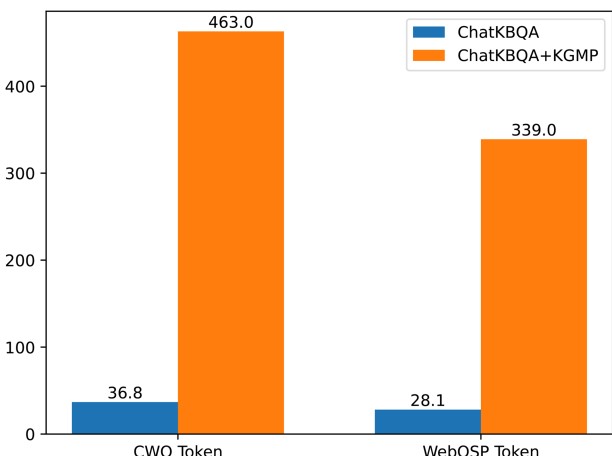

**Fig 6. Comparison of average input Tokens length between ChatKBQA and ChatKBQA+KGMP.**

Assuming a batch size of B and an input sequence length of N, we compute the FLOPs for the self-attention mechanism as follows:

**Linear Transformations:** Each transformation (query, key, value) involves $2 \times B \times N \times d_{model} \times d_k$ FLOPs, which adds up to $3 \times B \times N \times d_{model} \times d_k$ for all transformations.

**Attention Score Calculation:** The formula $(Q \cdot K^T)$ gives $2 \times B \times N \times N \times H \times d_k$ FLOPs.

**Application of Attention:** The formula $softmax(\frac{QK^T}{\sqrt{d_k}} \cdot V)$ gives $2 \times B \times N \times N \times H \times d_k$ FLOPs.

So the total FLOPs for the self-attention mechanism per layer is $FLOP_{self-attention} = 6 \times B \times N \times d_{model} \times d_k + 4 \times B \times N^2 \times H \times d_k$.

For the FFN component:

**First Linear Transformation:** Involves $2 \times B \times N \times d_{model} \times d_{ff}$ FLOPs.

**Second Linear Transformation:** Involves $2 \times B \times N \times d_{ff} \times d_{model}$ FLOPs.

Therefore the total FLOPs for each FFN layer is $FLOPs_{FFN} = 4 \times B \times N \times d_{model} \times d_{ff}$.

Summing the FLOPs of all the components and multiplying by the number of layers L gives the total FLOPs: $FLOPs_{total} = L \times (FLOP_{self-attention} + FLOPs_{FFN})$. Substituting the values into the formula, Table 2 shows that increasing the input sequence length leads to a rapid escalation in the number of FLOPs, highlighting the increased computational requirements of the KGMP.

**Subgraph mismatch.** The subsequent error analysis demonstrated that 44% of errors are attributable to incorrect path selection. The path selection errors mainly come from following three factors: (1) improper selection of the direction of high-order subgraph expansion; (2) wrong relationship selection due to semantic bias in the graph query language generation process even though the subgraph is correct; and (3) premature termination of the traversal process when the expected subgraph size is not reached. It is worth noting that when a subgraph direction selection error occurs, the model terminates the traversal after a fixed number of steps even if the current subgraph does not contain the target answer. This property may be related to the inherent properties of the datasets - the fine-tuned data of KGMP comes from two datasets of WebQSP and CWQ, which have significant differences in the size of subgraphs involved in multi-hop queries, and the average number of traversals of KGMP is not the same on two datasets when reasoning, which is closely related to the distributional characteristics of query complexity between different datasets.

In order to further verify the model robustness, this study constructs a test set containing error subgraphs as well as subgraphs without high-order for validation experiments. The experimental results show that KGMP fails to derive the correct answers from the error subgraphs or subgraphs without high-order, which is directly related to the generation mechanism of its training data. Since the training samples of the model are all generated based on completely correct raw data, KGMP fails to acquire the inference pattern of the error subgraph or subgraph without high-order during the training process, resulting in its inference ability based on the correct knowledge subgraph only. This finding reveals the limitations of current knowledge graph inference models in terms of fault-tolerance mechanisms and provides an important direction for future study.

**Table 2**. ChatKBQA and ChatKBQA+KGMP FLOPs with different tokens inputs.

| Datasets | Model | Average Input Token | FLOPs(GFLOPs) |
|---|---|---|---|
| CWQ | ChatKBQA | 36.8 | 30.96 |
| | ChatKBQA+KGMP | 463.0 | 402.36 |
| WebQSP | ChatKBQA | 28.1 | 24.23 |
| | ChatKBQA+KGMP | 339.0 | 292.12 |

**KGMP performance on very large KG.** In order to investigate the impact of knowledge graph size on KGMP performance, the database memory is constrained in order to simulate the situation of an extremely large knowledge graph being loaded. The query-answering time was split into two parts: (1) GQL execution time (largely affected by CPU traversal speed and the amount of available memory for indexing); and (2) LLM inference time (primarily running on GPU and not sensitive to CPU/memory constraints). As the size of the knowledge graph increased (i.e., the available memory decreased), a significant rise in GQL execution time was observed, leading to a substantial increase in overall query time. In contrast, LLM inference time remained almost constant throughout, primarily due to the fact that the reranker module prunes the candidate relations to a fixed number, thereby ensuring that the input to the LLM remains approximately the same size, regardless of the graph's overall scale. This phenomenon is further elucidated in Fig 7, which demonstrates that GQL execution time exerts a dominant influence on the overall performance cost in large graphs, while the LLM inference component remains relatively unimpacted.

**Retrieval performance degradation in long input sequences.** In KBQA, multi-hop inference problems often require traversing multiple subgraph levels, which leads to an exponential increase in the length of the input sequence. It is worth noting that Liu et al. [43] empirically found that there is a significant "Lost in the middle" phenomenon in the knowledge retrieval process of large-scale language models, which is manifested in the following two aspects: (1) Document position sensitivity: When the document containing the correct answer is located at the beginning or end of the input sequence, the model retrieval accuracy can reach the peak. However, if it is in the middle of the sequence, the accuracy decreases by more than 20%, which shows obvious retrieval position bias. (2) Sequence length dependence: In the control variable experiments, when the input sequence lengths are 2k, 4k and 6k tokens, the model's retrieval accuracy for the first document shows a decreasing trend (77% → 75% → 72%), indicating that long sequence processing exacerbates the model performance decay. This finding has important implications for KGMP: the linear growth of input sequences and the nonlinear decay of model performance form a sharp contradiction when current architectures extend the depth of subgraph search. Therefore, how to construct an efficient sequence compression mechanism to reduce computational complexity while

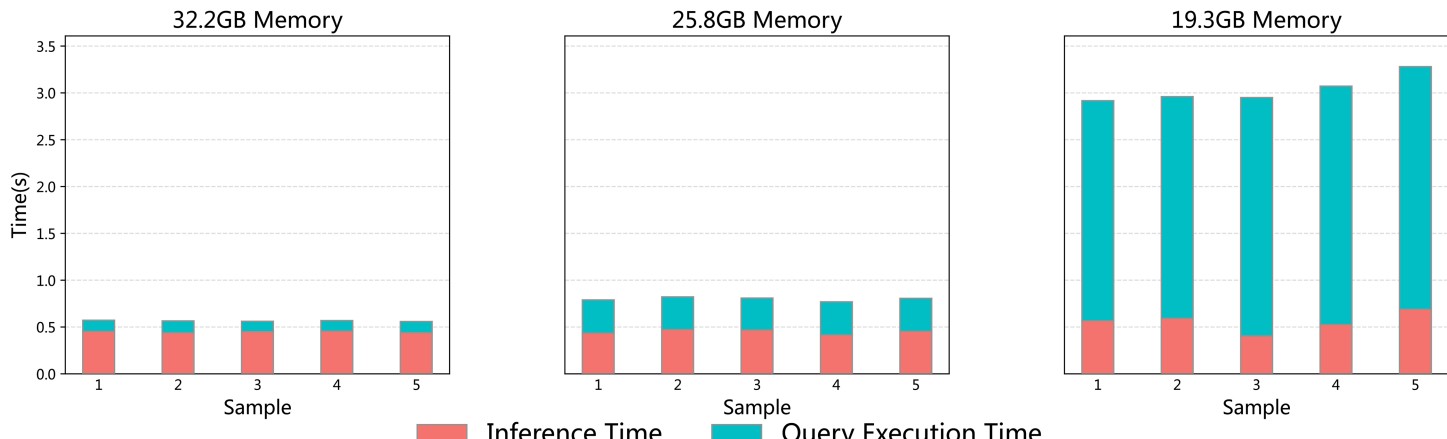

**Fig 7. Comparison of LLM inference elapsed time and database GQL execution elapsed time with different memory footprint reductions.**

maintaining semantic integrity will become a key research direction to improve the effectiveness of multi-hop inference.

## Determining subgraph information sufficiency in LLMs

In the polling mechanism of KGMP, the core challenge lies in how LLMs dynamically determine subgraph expansion based on user query semantics and current subgraph topology. This section investigates the formation mechanism of LLMs' decision logic, where the judgment capability primarily stems from two aspects of knowledge internalization: historical pattern transfer learning and knowledge-driven state evaluation.

**Historical Pattern Transfer Learning:** Through fine-tuning with complete query path samples (which consist of triplets representing a question, the corresponding subgraph state, and the expansion decision), LLMs acquire two critical decision patterns: (1) Initiate subgraph expansion when existing GQL query paths lack crucial semantic-associated edges; (2) Terminate expansion upon detecting minimal connected subgraphs sufficient for problem-solving.

**Knowledge-Driven State Evaluation:** LLMs' capability to assess subgraph information adequacy derives from dual knowledge integration: (1) **Pretrained Knowledge Base:** Massive corpus pretraining establishes commonsense knowledge repositories, enabling deep semantic parsing of natural language instructions and providing cognitive frameworks for domain knowledge transfer; (2) **Domain-Specific Fine-Tuning:** Through structured GQL syntax samples and logical reasoning chain reinforcement, LLMs develop mapping capabilities between graph topology and problem semantics. For instance, when processing the query "What electorate does Anna Bligh represent?", the model executes the following reasoning chain: (a) The primary subgraph contains only the attributes of the *government.politician entity* and lacks associations with electoral districts; (b) Activates pattern matching mechanism to select *government_positions_held relationship* for expansion; (c) Detects *district_represented* attribute edge in second-order subgraph, confirming information completeness; (d) Terminates expansion and generates final answer.

This hierarchical knowledge architecture enables LLMs to dynamically evaluate subgraph states and autonomously optimize query paths, with decision-making essentially representing synergistic computation between pretrained knowledge generalization and domain-specific fine-tuning specialization.

## Future directions

In the subsequent research, the optimization of KGMP will focus on two core directions. Firstly, the dual goals of optimizing computational power consumption and improving robustness will be achieved by focusing on breaking through the key technology of effective error correction when the system makes mistakes during subgraph traversal under low computational power constraints. Secondly, the performance limits under the scenario of ultra-large knowledge graphs will be investigated, and the robustness improvement mechanism will be explored based on reinforcement learning. It is noteworthy that the optimization of query strategies to minimize interaction frequency with the database can be an effective means of enhancing the execution efficiency of KGMP in billion-scale ternary knowledge graphs.

## Conclusions

Precise identification of subgraph depth present considerable challenges for existing KBQA methodologies. To address these issues, we propose the use of a retrieval-generation framework for KBQA named KGMP. KGMP employs fine-tuned LLMs and an iterative query strategy to effectively extract high-order subgraph information, thereby providing critical structural support for the generation of executable GQLs. The experimental evaluation is based on two standard KBQA datasets, WebQSP and CWQ. The results demonstrate that the incorporation of high-order subgraph information in KBQA through KGMP markedly enhances the precision of graph query generation by LLMs. Future research will investigate effective approaches for transferring and fine-tuning diverse knowledge graphs, building upon prior investigations in graph adaptation [52], as well as investigating ways to reduce the number of interaction rounds and the computational requirements.

## Author contributions

**Conceptualization:** Zhijie Yang, Liang Zhang, Duanbing Chen.

**Data curation:** Liyuan Weng, Ruojia Tong.

**Formal analysis:** Jianyu Xie, Zhuo Zeng.

**Funding acquisition:** Liyuan Weng.

**Investigation:** Zhijie Yang.

**Methodology:** Liang Zhang, Duanbing Chen.

**Project administration:** Liang Zhang, Ruojia Tong, Jianyu Xie, Zhuo Zeng.

**Resources:** Jianyu Xie, Zhuo Zeng.

**Software:** Zhijie Yang, Liang Zhang, Duanbing Chen.

**Supervision:** Zhijie Yang, Liyuan Weng, Liang Zhang, Duanbing Chen.

**Validation:** Ruojia Tong, Jianyu Xie, Zhuo Zeng.

**Visualization:** Jianyu Xie.

**Writing – original draft:** Zhijie Yang, Liang Zhang, Ruojia Tong, Duanbing Chen.

**Writing – review & editing:** Zhijie Yang, Liyuan Weng, Liang Zhang, Ruojia Tong, Jianyu Xie, Zhuo Zeng, Duanbing Chen.

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
