## [Decision Letter · Decision Letter 0]

4 Feb 2025

PONE-D-24-59861KGMP: Argumenting Retrieval Knowledge Graph with Multi-hop PerceptronPLOS ONE

Dear Dr. Chen,

Thank you for submitting your manuscript to PLOS ONE. After careful consideration, we feel that it has merit but does not fully meet PLOS ONE’s publication criteria as it currently stands. Therefore, we invite you to submit a revised version of the manuscript that addresses the points raised during the review process.

We look forward to receiving your revised manuscript.

Kind regards,

Ashraf Osman Ibrahim

Academic Editor

PLOS ONE

Journal Requirements:

5. Please upload a copy of Figure 5, to which you refer in your text on page 11. If the figure is no longer to be included as part of the submission please remove all reference to it within the text.

Reviewers' comments:

Reviewer's Responses to Questions

**Comments to the Author**

1. Is the manuscript technically sound, and do the data support the conclusions?

Reviewer #1: Yes

Reviewer #2: Yes

2. Has the statistical analysis been performed appropriately and rigorously? 

Reviewer #1: Yes

Reviewer #2: Yes

3. Have the authors made all data underlying the findings in their manuscript fully available?

Reviewer #1: Yes

Reviewer #2: Yes

4. Is the manuscript presented in an intelligible fashion and written in standard English?

Reviewer #1: Yes

Reviewer #2: Yes

5. Review Comments to the Author

Reviewer #1: This paper proposes the Knowledge Graph Multi-hop Perceptron (KGMP) module to enhance Knowledge-Based Question Answering (KBQA) systems by addressing the challenge of multi-hop questions. The proposed framework leverages fine-tuned open-source Large Language Models (LLMs) for retrieving and generating answers from knowledge graphs. A key innovation of KGMP is its ability to assess the sufficiency of the current subgraph information for transforming a natural language query into a Graph Query Language (GQL) and request higher-order subgraphs when necessary. Experimental results show significant improvements in accuracy on WebQSP and CWQ datasets, with increases of 6.2% and 5.3%, respectively, demonstrating the effectiveness of the approach. Additional validation is provided using models like StructGPT. There are some remaining issues:

While the proposed framework effectively addresses multi-hop queries, the mechanism for determining when additional subgraph information is needed could benefit from more detailed explanation.

The paper could elaborate more on potential limitations of the KGMP module, such as the scalability of the subgraph retrieval process and its impact on performance with very large knowledge graphs. Additionally, what happens when there is no higher-order subgraph available? Does the model still return an answer, or does it fail gracefully?

Missing references:

Think-on-graph 2.0: Deep and interpretable large language model reasoning with knowledge graph-guided retrieval

Retrieval-augmented generation with knowledge graphs for customer service question answering

OneGen: Efficient One-Pass Unified Generation and Retrieval for LLMs

Reviewer #2: Some sentences in the abstract were confusing, suggestion of comment attached in the reviewer file comment. Please add the study contributions and potential enhancement for future directions. Please also add discussion findings in the paper.

6. PLOS authors have the option to publish the peer review history of their article (what does this mean?). If published, this will include your full peer review and any attached files.

Reviewer #1: No

Reviewer #2: No

---

## [Author Response · Author response to Decision Letter 1]

2 Apr 2025

Dear Editor,

We have revised our manuscript according to the reviewers’ comments. Please find the detailed response to reviewers attached. Thank you for your valuable suggestions, and we look forward to your feedback on the revised version.

Yours Sincerely,

Duanbing Chen

---

## [Decision Letter · Decision Letter 1]

9 Sep 2025

KGMP: Augmenting Retrieval Knowledge Graph with Multi-hop Perceptron

PONE-D-24-59861R1

Dear Dr. Chen,

We’re pleased to inform you that your manuscript has been judged scientifically suitable for publication and will be formally accepted for publication once it meets all outstanding technical requirements.

Kind regards,

Sergio Consoli

Academic Editor

PLOS ONE

Additional Editor Comments (optional):

Reviewer #1:

Reviewer #3:

Reviewers' comments:

Reviewer's Responses to Questions

**Comments to the Author**

1. If the authors have adequately addressed your comments raised in a previous round of review and you feel that this manuscript is now acceptable for publication, you may indicate that here to bypass the “Comments to the Author” section, enter your conflict of interest statement in the “Confidential to Editor” section, and submit your "Accept" recommendation.

Reviewer #1: All comments have been addressed

Reviewer #3: (No Response)

2. Is the manuscript technically sound, and do the data support the conclusions?

Reviewer #1: Partly

Reviewer #3: Yes

3. Has the statistical analysis been performed appropriately and rigorously? 

Reviewer #1: Yes

Reviewer #3: Yes

4. Have the authors made all data underlying the findings in their manuscript fully available?

Reviewer #1: No

Reviewer #3: Yes

5. Is the manuscript presented in an intelligible fashion and written in standard English?

Reviewer #1: Yes

Reviewer #3: Yes

6. Review Comments to the Author

Reviewer #1: This study proposes Knowledge Graph Multi-hop Perceptron (KGMP) - a retrieval-generation framework fine-tuned based on open-source large language models. The revised version has addressed my concerns.

Reviewer #3: The paper is well-written and tackles an important problem in KBQA.

The proposed framework is clearly described and experimentally validated.

The comparison with existing literature is thorough and useful.

Results show promising improvements over strong baselines.

The work has potential impact in advancing LLM–KG integration.

A few areas (e.g., broader datasets, error analysis, generalizability) could be expanded in future versions

7. PLOS authors have the option to publish the peer review history of their article (what does this mean?). If published, this will include your full peer review and any attached files.

Reviewer #1: No

Reviewer #3: **Yes: **Dalin Almbaidin

---

## [Editor Report · Acceptance letter]

PONE-D-24-59861R1

PLOS ONE

Dear Dr. Chen,

I'm pleased to inform you that your manuscript has been deemed suitable for publication in PLOS ONE. Congratulations! Your manuscript is now being handed over to our production team.

Kind regards,

on behalf of

Dr. Sergio Consoli

Academic Editor

PLOS ONE